# Astrocytes in Rodent Anxiety-Related Behavior: Role of Calcium and Beyond

**DOI:** 10.3390/ijms26062774

**Published:** 2025-03-19

**Authors:** Marta Gómez-Gonzalo

**Affiliations:** Section of Padua, Neuroscience Institute, National Research Council (CNR), 35131 Padua, Italy; marta.gomezgonzalo@cnr.it

**Keywords:** astrocyte, anxiety behavior, mental health, calcium signaling, chemogenetic stimulation, optogenetic stimulation, knockout mice

## Abstract

Anxiety is a physiological, emotional response that anticipates distal threats. When kept under control, anxiety is a beneficial response, helping animals to maintain heightened attention in environments with potential dangers. However, an overestimation of potential threats can lead to an excessive expression of anxiety that, in humans, may evolve into anxiety disorders. Pharmacological treatments show variable efficacy among patients, highlighting the need for more efforts to better understand the pathogenesis of anxiety disorders. Mounting evidence suggests that astrocytes, a type of glial cells, are active partners of neurons in brain circuits and in the regulation of behaviors under both physiological and pathological conditions. In this review, I summarize the current literature on the role of astrocytes from different brain regions in modulating anxious states, with the goal of exploring novel cerebral mechanisms to identify potential innovative therapeutic targets for the treatment of anxiety disorders.

## 1. Introduction

Anxiety is a common emotional response that helps animals anticipate potential threats and maintain high levels of attention in environments with perceived dangers. In this context, anxiety plays a pivotal role in driving avoidance behaviors in animals. While mild levels of anxiety can be ethologically relevant for the survival of individuals, the overestimation of perceived threats can lead to excessive anxiety, resulting in maladaptive and disproportioned avoidance behaviors. As a result, “states” of controlled anxiety contrast with “traits” of pathological anxiety [1], which characterize the various anxiety disorders in humans: generalized anxiety disorder, panic disorder with or without agoraphobia, specific phobias, agoraphobia, social anxiety disorder, separation anxiety disorder and selective mutism [2]. A study using data from 204 countries between 1990 and 2019 concluded that anxiety disorders are the most common mental illness in the world, with a significant impact on the global burden of disease [3]. These debilitating conditions are multifactorial in origin, with significant contributions from social, psychological and biological factors [4]. Additionally, the familiarity observed in some cases of anxiety disorders suggests a potential genetic contribution [5].

Despite the evident difficulties in using animal models to uncover the biological basis of the highly subjective emotional states involved in some human behaviors/pathologies [6], significant efforts have been made to dissect out the complex mechanisms underlying anxiety behavior in rodent studies. Among the rodent behavioral tests used to investigate anxiety-like levels in animals, approach–avoidance tests are the most common paradigms employed [7]. In these tests, the anxiety-like behavior of a rodent is measured by the conflict created when a natural approach behavior is opposed to a relevant avoidance behavior. Natural approach behaviors include actions that drive the search for natural rewards, such as exploration of new contexts and engaging in social interactions. These approach behaviors are measured in environments that may be perceived as potentially threatening, such as open or brightly lit spaces, and the quantitative assessment of the approach behavior can be interpreted as an index of the anxiety level. These approach–avoidance tests have been described in detail elsewhere [1,8], and some of them have been validated across species [9]. For a better comprehension of the studies discussed in this review, I briefly describe the most relevant tests here (Figure 1). The open field test (OFT) involves positioning the rodent in a novel environment, i.e., an unescapable arena, where its exploratory behavior of the novel environment (horizontal and vertical activity), overall locomotor activity (total distance traveled) and the anxiety-related behavior (time spent in the anxiogenic center of the arena relative to the time spent in the safer proximity of the walls) can be assessed. The elevated plus maze (EPM) test consists of placing the rodent in an elevated cross-shaped structure with two open arms and two wall-protected arms. In this test, the time spent in the more anxiogenic open arms and the number of entries in them are reduced in animals with higher anxiety levels. A variant of the EPM test is the zero maze test, in which two wall-protected sections and two exposed sections are alternately arranged along a circular tract. Finally, the light–dark box (LDB) test involves placing the animal in an arena with two compartments, one brightly lit and open and the other completely dark and enclosed. In this test, the time spent in the more anxiogenic light compartment and the number of entries into it decreases as anxiety levels increase.

In humans, stressful conditions favor and reinforce anxiety-related disorders, and chronic stress can contribute to depression [10]. In mental health consultations, one of the most common comorbidities observed is the co-occurrence of anxiety and depression disorders [11]. Patients with anxiety disorders are at higher risk for developing depression, and although these two psychiatric conditions are clinically distinct, they share many overlapping symptoms. Another human psychiatric disorder that is commonly comorbid with anxiety disorders is the post-traumatic stress disorder (PTSD) [12]. PTSD, classified by the DSM-5 at present as a trauma and stressors-related disorder, and no longer as an anxiety disorder as it was previously, is characterized by the experience of a catastrophic event that can be either emotional or physical. These disorders highlight the critical role that stressors, ranging from mild forms to more severe traumatic events, play in the onset of mental health difficulties when individuals fail to cope with adverse conditions.

Laboratory research on anxiety behavior in rodents has not been limited to investigating basal anxiety-like levels. Similarly to humans, the exposure of rodents to different types of stressors can increase anxiety levels. Various stress models in rodents, such as chronic stress models and developmental stress models, have been designed to understand the effects of stress on anxiety levels (for a detailed description, see [7]). Although it has been argued that the forced expression of anxiety in rodents could still fall within the normal range of adaptive coping responses and that this experimental approach does not fully achieve the ambitious goal of modeling clinical symptoms in humans [13], most stress-related models in rodents result in the comorbid expression of elevated anxiety levels along with depressive symptoms or fear manifestation. This observation mirrors the comorbid expression of human anxiety disorders with other psychiatric pathologies, supporting the use of stressors in the study of anxiety-related behaviors.

Astrocytes are glial cells that “sense” ongoing levels of neuronal activity through receptors for common neurotransmitters and neuromodulators on their membranes, including glutamate, GABA, ATP, D-serine, acetylcholine, dopamine, serotonin, histamine, endocannabinoids and norepinephrine [14]. Activation of these receptors results in cytosolic Ca^2+^ transients in astrocytes, both at the soma and in fine processes intimately in contact with neuronal synapses [15,16]. Astrocyte Ca^2+^ elevations elicited by neuronal mediators crucially depend on the activation of various G protein-coupled receptors (GPCRs), which are associated with diverse intracellular signaling pathways [14,17,18]. Astrocyte activation ultimately modulates neuronal excitability and circuits through the release of gliotransmitters, alterations in the activity levels of neurotransmitter transporters and/or variations in the extracellular concentration of ions [19,20,21]. This modulation of neuronal circuits by astrocytes ultimately results in behavioral changes, and it is currently thought that information processing in the brain depends on a network of highly interacting neurons and astrocytes [22]. Given the frequent use of chemogenetic approaches to manipulate astrocyte activity, I will briefly describe the main designer receptors exclusively activated by designer drugs (DREADDs) that have been used to manipulate astrocytes in anxiety research. Like the GPCRs endogenously expressed by astrocytes, DREADDs are engineered GPCRs associated with diverse intracellular signaling pathways [18,23] after activation with designer drugs. The two receptors more frequently used in the field of anxiety are hM3Dq and hM4Di, coupled to Gq or Gi alpha subunits, respectively, and are commonly activated with clozapine N-oxide (CNO). Canonically, Gq-DREADDs activate phospholipase C (PLC) to produce diacylglycerol (DAG) and inositol 1,4,5-triphosphate (IP3) as second messengers. It is well known that Gq–GPCRs in astrocytes trigger the release of Ca^2+^ from the endoplasmic reticulum (ER), mainly through the activation of IP3-sensitive receptors with Ca^2+^ channel activity (IP3Rs) [18]. In contrast, Gi-DREADDs are typically associated with the inhibition of adenylyl cyclase activity and the ensuing reduction in the intracellular levels of cAMP. This signaling pathway canonically result in the inhibition of neurons. However, the inhibition of astrocytes by Gi-DREADDs remains a controversial issue. Interestingly, the activation of Gi-GPCRs and Gi-DREADDs in astrocytes can also trigger Ca^2+^ transients through non-canonical IP3R activation mechanisms [22,24,25,26,27,28]. However, a study from Goshen’s lab described that the activation of Gi signaling in astrocytes may curiously exert a negative impact on Ca^2+^ signaling [29]. Although the cellular mechanism responsible for the interference with Ca^2+^ levels/signaling has not been elucidated, these studies overall illustrate the complexity of Ca^2+^ signaling regulation in astrocytes and raise concerns about the use of Gi-DREADDs to “inhibit” Ca^2+^ events in astrocytes, highlighting the need to validate these tools in each individual experiment. Despite advances made in the field, this is often overlooked in behavioral studies due to the technical complexity of studying Ca^2+^ in vivo. As a result, in those behavioral studies where in vivo Ca^2+^ signaling is not investigated alongside the behavioral output, the role of astrocyte Ca^2+^ signaling cannot be unambiguously established when activating Gi signaling. Although Ca^2+^ signaling in astrocytes has been extensively studied and remains a major intracellular signaling pathway during astrocyte function [30,31], it is not the only intracellular signaling pathway in these glial cells. In recent years, significant efforts have been made to investigate how cAMP impacts astrocyte functions [32]. In this review, it will become clear that boosting Gq or Gi signaling in astrocytes often leads to bidirectional modulation of anxiety levels. While this could be explained by distinct effects on astrocyte Ca^2+^ signaling, it is tempting to speculate that this bidirectional control may stem from the regulation of distinct intracellular signaling pathways within astrocytes.

Regarding Gq signaling activation by DREADDs in astrocytes, a critical issue to consider in these studies is the timing of the designer drug administration to activate DREADDs in relation to the timing of the behavioral test. In the study conducted by Vaidyanathan et al. in cortical astrocytes, the authors demonstrated that the in vivo Gq-mediated chemogenetic activation of astrocytes increased their Ca^2+^ dynamics, in terms of frequency of Ca^2+^ elevations, observed within the first 30 min after the intraperitoneal (i.p.) injection of CNO in a dose-dependent manner [28]. This increase was followed by a dramatic decrease in Ca^2+^ dynamics, caused by a sustained high Ca^2+^ plateau that lasted for at least one hour. Given that astrocyte responses characterized by oscillatory Ca^2+^ signaling dynamic may have functional effects more relevant than just sustained plateau increases [33], a theoretical biphasic response could complicate the interpretation of the results obtained from behavioral tests performed 30 min after chemogenetic activation of astrocytes. With Ca^2+^ dynamics similar to those evoked by Gq-DREADD activation in cortical astrocytes, exogenous activation of Gq signaling in astrocytes might, paradoxically, interfere with the endogenous Gq signaling activation and the precise spatiotemporal Ca^2+^ activity required during natural behavior, leading to unexpected behavioral outcomes.

According to the points mentioned previously, there are two concepts that should always be considered when investigating the involvement of astrocytes in animal behavior. With the advent of new tools to activate astrocyte Ca^2+^ signaling, a tempting belief has emerged that establishing a causal relationship between astrocyte activity and behavior can always be achieved by using any of the currently available methods to exogenously activate astrocytes. However, accurately mimicking endogenous astrocyte activity is a challenging task, and the different spatiotemporal features elicited by various activation methods, often far from the endogenous activity associated with natural behavior, can lead to disparate behavioral effects (for an example, see the “Hippocampus” chapter in this review). This suggests that the different tools available for astrocyte activation, along with the specific experimental protocols applied, can activate a variety of intracellular signaling pathways within astrocytes, ultimately leading to different behavioral outcomes. In other words, the behavioral consequences observed after exogenous astrocyte activation reflect the “potential effects” that astrocyte activity can exert on that particular behavior, but these effects may not necessarily align with the “real effects” of endogenous astrocyte activity on that behavior. This apparent limitation, however, can become a valuable tool for manipulating neuronal circuits in different ways. In contrast, if the goal of an experiment is to establish unambiguous causal relationships between astrocyte Ca^2+^ signaling and behavioral outcome, tools that impair astrocyte function during natural behavior should be preferred over those that boost astrocyte activity.

Approach–avoidance paradigms have been used to investigate the effectiveness of anxiolytic treatments for human disorders ([2,9], but see also [34]) and to explore the role of specific cerebral circuits in anxiety-like behaviors in rodents. Anxiety, like other complex behaviors, depends on a network of cellular communication within and between different brain circuits [2,35,36]. This review, which presents the most recent discoveries on the role of astrocytes in anxiety-like behavior in rodents, is organized into chapters, each focusing on a different brain circuit involved in anxiety. Although I will place special emphasis on the pivotal role of astrocyte Ca^2+^ signaling in modulating anxiety levels, astrocytic Ca^2+^-independent mechanisms will also be discussed. Figure 2 schematically summarizes the astrocyte observations and manipulations associated with anxiolytic and anxiogenic effects.

## 2. Hippocampus

The hippocampus is a complex and highly organized brain circuit characterized by functional heterogeneity along the dorsoventral axis, with the dorsal hippocampus being critical for episodic memory and spatial navigation and the ventral hippocampus implicated in the processing of affective states. The hippocampal neurons coding for the mental representation of external space, known as “place cells”, are enriched in the dorsal hippocampus, while “anxiety neurons” have been described as preferentially located in the ventral hippocampus [37]. Anxiety neurons are ventral CA1 neurons that increase their activity in anxiogenic environments, such as the open arms of the EPM [37]. Activation of hippocampal anxiety–neuron projections to the lateral hypothalamus, but not to the basal amygdala, increases anxiety levels and reinforces avoidance behaviors [37]. Likewise, direct projections from the ventral hippocampus to the medial prefrontal cortex promote the expression of anxiety-like behavior [38]. In accordance with the presence of anxiety-neurons in the ventral hippocampus, lesions of the ventral, but not dorsal, hippocampus yield anxiolytic effects [39,40]. The CA1 is not the only hippocampal subregion participating in anxiety-like behavior. Granule cells from ventral dentate gyrus also modulate this affective behavior, as shown by the anxiolytic effect elicited by the activation of these hippocampal cells, which leads to a powerful suppression of innate anxiety [41].

A recent research study performed by Cho et al. in the dorsal hippocampal region of male mice found that also astrocyte activity modulates anxiety-like behavior [42]. The authors performed two-photon in vivo Ca^2+^ imaging in mice with tamoxifen-inducible expression of the Ca^2+^ indicator GCaMP6s, specifically in astrocytes. In head-fixed, awake mice, the exploration of a virtual reality environment simulating two closed arms separated by an open center revealed that the Ca^2+^ levels of astrocytes located in the stratum radiatum and stratum lacunosum–molecolare of the dorsal hippocampus were increased while mice explored the anxiogenic open center of the virtual reality environment. Similarly, in separate experiments in which a black or bright white whole screen was alternatively presented to the mice, an increase in astrocyte Ca^2+^ activity was observed when switching to the anxiogenic bright screen. These results suggest a correlation between astrocyte Ca^2+^ activity and anxiogenic environments. To explore the “causal” link between astrocyte Ca^2+^ activity and anxiety-like behavior (see the “Introduction”), the authors used an optogenetic approach to increase Ca^2+^ signals in astrocytes while assessing anxiety-like behavior in the EPM test and locomotion behavior in the OFT. In particular, the authors used the blue light-gated channel Channelrhodopsin-2 (ChR2), which causes cellular depolarization by non-selective cation flow (including Na^+^, H^+^, K^+^ and Ca^2+^) through the plasma membrane. Although this tool has proven highly valuable in neuronal studies, it is an unusual method for activating astrocytes, as the Ca^2+^ increases evoked by this method do not engage the signaling pathways naturally employed by astrocytes [31]. After tamoxifen-inducible expression of ChR2 in hippocampal astrocytes, light stimulation of the dorsal hippocampus induced anxiolytic effects, as revealed by an increase in the exploration time in the open arms, along with an increase in the exploratory behavior and locomotor speed in the OFT. Interestingly, optogenetic stimulation of ventral hippocampal astrocytes yielded similar anxiolytic effects in the EPM test but, in this case, without changes in the locomotor speed in the OFT. Taken together, these results suggest that optogenetic stimulation of hippocampal astrocytes exerts an anxiolytic effect regardless of the dorsoventral axis. Optogenetic activation of dorsal or ventral hippocampal astrocytes led to an increased expression of neuronal c-Fos (a marker of cellular activation) mainly in DG, as well as in CA1 and CA3. Interestingly, the increase in c-Fos occurs simultaneously in both dorsal and ventral hippocampal neurons, regardless of the dorsoventral position of the activated astrocytes. Excluding a technical pitfall [6], this result suggests that, despite the functional heterogeneity along the dorsoventral axis, there is a complex interplay, possibly mediated not only by neurons but also by astrocytes, between dorsal and ventral hippocampus.

Authors then moved to brain slice electrophysiological experiments to elucidate the molecular mechanism underlying the anxiolytic effect of hippocampal astrocyte activation. In slices from both dorsal and ventral hippocampus, the authors found that the optogenetic activation of astrocytes increased the excitatory transmission into DG and CA1 neurons. This effect occurred through a mechanism that depends on the activation of receptors for the neuroactive molecule ATP, the latter likely released by astrocytes, as the ATP receptor inhibitor PPADS blunted the increased activity. The causal role of ATP in the anxiolytic effect of hippocampal astrocyte activation was supported by in vivo pharmacological experiments in which the ATP receptor signaling was locally hampered. In these experiments, the anxiolytic effect elicited by optogenetic stimulation of dorsal hippocampal astrocytes, in terms of time spent in the open arms, was impaired in the presence of the general ATP receptor inhibitor PPADS.

Although the correlation between reinforced astrocyte Ca^2+^ signaling in hippocampus and anxiogenic environments seems convincing, some issues must be considered in the experiments establishing a causal role between astrocyte Ca^2+^ signals and the modulation of anxiety levels. First, during optogenetic stimulation of astrocytes, the increase in extracellular K^+^ levels that accompanies ChR2 activation has raised concerns about the interpretation of the results obtained with this type of astrocyte stimulation, as increases in extracellular K^+^ may constitute a depolarizing stimulus for adjacent neurons (for details, see [43]). The authors addressed this concern by studying the effect of increasing the extracellular K^+^ levels on the excitatory transmission in hippocampal brain slices. They found that increasing the concentration of K^+^ in the bath perfusion did not mimic the synaptic changes elicited by optogenetic astrocyte stimulation, suggesting that, apart from K^+^, other mechanisms may drive the anxiolytic effects of astrocyte activation. Second, during optogenetic stimulation of astrocytes in the dorsal hippocampus, an increase in locomotion accompanied the observed anxiolytic effect. When anxiety levels are investigated using approaches based in the exploratory activity of mice, special caution should be taken regarding simultaneous effects on locomotor activity. While it is widely accepted that a manipulation yielding reduced locomotion could confound the assessment of anxiety levels [8,44], the increase in locomotion observed in the study by Cho et al. [42] should also be considered to avoid confounding interrelated behavioral effects. This concern was resolved, at least in part, during the in vivo experiments blocking ATP signaling. In these experiments, while the anxiolytic effect was significantly impaired, the potentiation of locomotor activity in response to astrocyte activation was highly preserved, suggesting that in the dorsal hippocampus, the two behavioral effects are not confounded.

Overall, this study reveals that hippocampal astrocytes are activated in anxiogenic environments and, given that the optogenetic activation of these glial cells exerts an anxiolytic effect, the authors hypothesized that the activation of hippocampal astrocytes in anxiogenic environments may be a physiological response to cope with potentially threatening conditions. This way, hippocampal astrocytes could be the target of novel therapeutic strategies to treat anxiety disorders. However, a subsequent study by Li et al. has provided results that do not support this hypothesis [45]. In contrast, the authors propose that hippocampal astrocyte activation has indeed an anxiogenic effect. This study, conducted in awake mice during spatial exploration in the EPM test and in the OFT, focused their research on ventral hippocampal astrocytes from male mice. These mice expressed the Ca^2+^ indicator GCaMP6m using an adeno-associated virus (AAV) strategy, and the mean Ca^2+^ levels of a population of astrocytes located in the ventral hippocampus was monitored during the exploratory activity with a fiber photometry approach. In accordance with the data presented by Cho et al. [42], the astrocyte population from ventral hippocampus responded with a Ca^2+^ elevation during the approach of mice to the naturally anxiogenic open arms of the EPM and the center region in the OFT [45]. Interestingly, the onset of the astrocyte responses occurred immediately before mice entered the anxiogenic area. Subsequently, the authors showed a bidirectional astrocyte regulation of anxiety levels that, however, was in contrast to the anxiolytic effects of hippocampal astrocyte activation described by Cho et al. [42]. To investigate the causal role of the astrocyte Ca^2+^ signaling during anxiety-like behavior, Li et al. generated knockout mice for the gene encoding the IP3 receptor type-2, the main endoplasmic receptor that mediates Ca^2+^ elevations in astrocytes [15,16,27,46], using an AAV strategy to express the recombinase Cre in hippocampal astrocytes from IP3R2^flox/flox^ mice. In these mice, the Ca^2+^ levels of hippocampal astrocytes did not increase when mice explored the open arms of the EPM. Interestingly, in these mice, but not in mice knocked out for the IP3R2 specifically in neurons, the time spent exploring anxiogenic areas (center region in the OFT and open arms in EPM) increased without changes in the traveled distance in both tests, suggesting that limiting the Ca^2+^ signaling in ventral hippocampal astrocytes exerts an anxiolytic effect in the innate levels of anxiety displayed by mice during exploratory activities. In accordance with this hypothesis, but in apparent contrast with the study by Cho et al. [42], chemogenetic activation (30 min before behavioral tests) of ventral hippocampal astrocytes expressing the Gq-DREADD receptor hM3Dq decreased the time spent in less anxiogenic areas, without changes in the general locomotor activity in both tests, suggesting that chemogenetic activation of ventral hippocampal astrocytes exerts an anxiogenic effect during the exploratory activity of mice. Although the anxiogenic effects observed after Gq-chemogenetic activation of astrocytes are consistent with the anxiolytic effects observed when the Ca^2+^ signaling in ventral hippocampal astrocytes is compromised, in vivo Ca^2+^ imaging experiments with Gq-mediated chemogenetic activation are needed to unambiguously establish the anxiogenic role of putative Ca^2+^ increases in hippocampal astrocytes.

In brain slices, the authors investigated the putative molecular mechanism of the anxiogenic effect of Ca^2+^ boosting and found that the Gq-chemogenetic activation of astrocytes increases the extracellular levels of glutamate and favors the tonic glutamatergic currents mediated by the NMDAR in ventral CA1 neurons. The causal effect of the activation of the NMDAR on the anxiogenic effect observed in response to astrocyte chemogenetic stimulation was supported by the use of NMDAR antagonists in vivo. Indeed, in the presence of agents locally impairing the NMDAR function in ventral hippocampus, the anxiogenic state induced by chemogenetic activation of astrocytes was ameliorated, supporting the hypothesis of the involvement of the NMDAR in the anxiogenic mechanism.

Authors finally observed that, compared to naive mice, mice subjected to a previous subacute restraint stress showed a higher level of anxiety in the OFT and EPM. Interestingly, the restraint stressor per se increased the glutamate levels in hippocampus without the participation of astrocytes, and in stressed mice, the activation of NMDARs also played a crucial role in the anxiogenic mechanism triggered by stress. Notably, compared to naive mice, stressed mice displayed a higher increase in the astrocyte Ca^2+^ signaling during the exploration of the anxiogenic open arms of the EPM, but a similar increase during the exploration of the center in the OFT, a less anxiogenic environment compared to the EPM. Then, the authors observed that, in mice subjected to subacute restraint stress, the manipulation of Ca^2+^ signaling in ventral hippocampal astrocytes yielded results consistent with the effects observed in naive mice. In stressed IP3R2 knockout mice, the impairment of astrocyte Ca^2+^ signaling conferred a reduction in the anxiety levels to an extent similar to that observed in naive mice, while Gq astrocyte activation worsened the anxiety-like behavior of stressed mice to an even higher extent than that observed in naive mice after Gq astrocyte activation.

The studies by Cho et al. [42] and Li et al. [45] propose disparate hippocampal astrocyte effects, with optogenetic activation exerting an anxiolytic effect, while Gq-related chemogenetic activation induces an anxiogenic effect. The apparently contrasting results can be reconciled by taking into account that the different tools used to activate hippocampal astrocytes activated distinct intracellular signaling pathways within astrocytes, which ultimately yielded different neuronal effects mediated by P2Rs or NMDARs, respectively.

Anxiety is a frequently observed symptom in patients with chronic pain [47]. The study conducted by Lv et al. investigated the role of hippocampal astrocytes in the anxiodepression associated with a mouse model of trigeminal neuralgia, in both male and female mice [48]. Fourteen days after chronic constriction of the infraorbital nerve (CION), mice displayed mechanical allodynia along with an increased anxious state, as revealed by a decrease in the time spent in the open arms of the EPM test, and a reduced number of open arm entries, without motor impairments in the OFT. Using different techniques (microdialysis and in vivo fiber photometry with genetically encoded adenosine sensors), the authors found that this model of trigeminal neuralgia was associated with an increased level of adenosine in the ventral CA1 hippocampal region. Interestingly, the application of an antagonist for the adenosine A2A receptor in the ventral CA1, but not an antagonist for the A1 receptor, significantly ameliorated the anxious state in CION mice. The use of an AAV strategy to target the expression of short hairpin RNAs (shRNAs) specifically to knock down the expression of A2A receptors in CA1 pyramidal neurons similarly attenuated the anxiety like-behavior without affecting mechanical allodynia. Adenosine is an enzymatic degradation product of ATP, a molecule known to be released by astrocytes through different pathways, including Cx43 hemichannels [49]. In CION mice, authors found that application of a Cx43 mimetic blocker peptide in the ventral CA1 reduced the extracellular levels of adenosine. Similarly, chemogenetic activation of the Gi signaling in ventral hippocampal astrocytes expressing hM4Di decreased the adenosine levels in the extracellular dialysate and, most importantly, improved the anxiety-related behavior without affecting mechanical allodynia. Although the authors did not check the Ca^2+^ signaling in vivo after CNO treatment, they observed reduced intracellular Ca^2+^ levels after application of CNO in hM4Di astrocytes from ventral hippocampal slices. The authors also provided evidence of the role of the ectonucleotidase CD39 expressed by microglia in the degradation of astrocyte-released ATP to adenosine. Considering additional data obtained from brain slices, the authors hypothesized that IL17 released by activated microglia could promote astrocyte activation and the release of ATP. The pivotal role of microglia in the astrocyte-mediated anxiety increase was corroborated by using different approaches to inhibit microglia. Indeed, when microglia were inhibited, anxiety behavior was attenuated without changes in mechanical allodynia. Finally, in brain slices, the authors found that CION increased the excitability of ventral CA1 pyramidal neurons, a characteristic of elevated anxiety states. The proposed mechanism of pain-mediated anxiety was further corroborated by experiments in brain slices, where the hyperexcitability of ventral CA1 pyramidal neurons was blocked by inhibiting the A2A receptor, the ectonucleotidase CD39 or microglia cells. Overall, these results, along with those by Cho et al. [42], elegantly illustrate how astrocytes integrate into functional units with different cellular and molecular characteristics that yield different brain outputs (in this case, anxiolytic or anxiogenic effects) in response to the same astrocyte action (in this case astrocyte ATP release, accompanied or not by ectonucleotidase activation).

Anxiety is a symptom that can also be associated with fear [36], and the hippocampus plays a critical role in the encoding of contextual fear memory [50]. In the study conducted by Li et al. [51], the authors observed that male rats, which associated foot shocks with specific cues or spatial contexts, also develop fear-related anxiety states, as indicated by reduced exploration of the center zone in the OFT (notably, the total distance traveled in this test was also reduced, although this concern was not discussed by the authors). Importantly, optogenetic activation of dorsal hippocampal astrocytes expressing ChR2, immediately after the fear conditional paradigm, reduced the context-mediated, but not cue-mediated, fear memory consolidation, as well as anxiety-like behavior. Photostimulation of hippocampal astrocytes increased extracellular levels of ATP and adenosine, and in contrast to the findings reported by Cho et al. [42], in which the anxiolytic effect observed by photoactivation of astrocytes was mediated by ATP receptors, the authors provided evidence that activation of the adenosine A1 receptor (A1R) reduced fear memory consolidation and exerted an anxiolytic effect on fear-related anxiety. These results highlight hippocampal astrocytes and A1Rs as potential new targets for the treatment of fear-related disorders, including fear-related anxiety.

While most studies activating astrocytes to observe changes in animal behavior used acute protocols for astrocyte activation, a recent study investigated the effects of chronic Gq activation of ventral hippocampal astrocytes on male mouse behavior [52]. The protocol used involved the addition of water-soluble deschloroclozapine, a drug designed to activate Gq-hM3Dq DREADD, to the home cage water supply for three, six or nine months. This unusual protocol did not change either the number of astrocytes and microglia or the overall morphology of these glial cells. Moreover, the cell territory of astrocytes remained unchanged among the different treatment durations, and only the cell territory of microglia was increased with chronic Gq astrocyte activation for three months. When authors conducted experiments to test anxiety levels in mice treated for different lengths of time, only mice treated for three months showed a reduced time spent in the center of the open field, without changes in the zero maze test. These results suggest that chronic activation of Gq astrocytes slightly affected the anxiety state, but only at the earliest time point investigated.

## 3. Prefrontal Cortex

The medial prefrontal cortex (mPFC), composed of the anterior cingulate (AAC), prelimbic (PL) and infralimbic (IL) cortex in rodents (for a discussion of anatomical homologs across humans and rodents, see [53]), is an important associative cortical region involved in higher brain functions, including the regulation of emotion processing during stress and anxiety [54,55]. The study conducted by Wu et al. investigated the Ca^2+^ signaling of astrocytes in the mPFC of male mice exposed to anxious environments (EPM, elevated platform) or negative stimuli (hanging by tail, foot shock, noxious olfactory cue and social defeat) [56]. The authors used an AAV strategy to express the Ca^2+^ sensor GCaMP6s in astrocytes from the IL region and employed fiber photometry to record mean changes in the fluorescence signal, reflecting Ca^2+^ changes in the “bulk” astrocyte population. Similarly to the Ca^2+^ elevations detected in hippocampal astrocytes when mice explored anxiogenic environments [42,45], mPFC astrocyte Ca^2+^ levels increased when mice first entered an open arm, and returned to low levels when they entered the close arms. This response was observed in subsequent explorations of the open arms, but with lower peak values. When mice were placed in an elevated platform, an anxiogenic environment due to its height, the Ca^2+^ signal in mPFC astrocytes rose each time the mouse’s head moved out of the platform. Likewise, all other stressors tested triggered Ca^2+^ peaks in mPFC astrocytes. These results suggest that PFC astrocytes respond to stressful and anxiogenic conditions.

Astrocytes from different brain regions respond with Ca^2+^ elevations to the activation of the GABA_B_ receptor [25,27,57,58]. A recent paper by Perea’s research team studied the functional consequences of the GABAergic signaling in mPFC astrocytes [59]. To achieve this goal, the authors used a Cre-dependent AAV strategy to delete GABA_B_ receptors specifically in astrocytes of the mPFC. Interestingly, male and female mice with downregulated expression of GABA_B_ receptors in these astrocytes exhibited lower anxiety levels, as revealed by increased time spent in the center of the OFT (without changes in average speed during movement) and a greater number of entries into the open arms of EPM test. These mice also exhibited deficits in mPFC-mediated memory processing, which were rescued by expressing and activating the light-activated GPCR melanopsin in PFC astrocytes. Although the authors provided evidence for the mechanism underlying the involvement of astrocyte GABA_B_ receptors in memory processing, they unfortunately did not further investigate whether the memory deficits and anxiolytic effects shared a similar mechanism, nor whether melanopsin could restore normal anxiety levels.

Stressful conditions, known to exacerbate anxiety symptoms, typically activate the hypothalamic–pituitary–adrenal (HPA) axis and lead to the concomitant release of glucocorticoid hormones (cortisol in humans and corticosterone in rodents), which in turn activate the glucocorticoid receptor (GR) [60]. GR is a transcription factor expressed in various cell types in the brain, including astrocytes [61]. A simple and commonly used mouse model to investigate stress in anxiety and depressive behavior is corticosterone treatment. In a study performed by Perea’s group, male mice were treated with a chronic, decreasing concentration of corticosterone in their daily water supply [62]. In juvenile mice with chronic oral corticosterone consumption, the authors found deficits in mood behavior, including increased depressive and despair-like behaviors, reduced sociability and heightened anxiety levels in the EPM test. The combined use of fiber photometry and an AAV strategy to express GCaMP6f in PFC astrocytes revealed an abnormal increase in Ca^2+^ signaling in vivo, in terms of velocity, amplitude and frequency of spontaneous Ca^2+^ elevations in corticosterone-treated mice exploring an open field. However, during social interaction in the three-chamber social test, Ca^2+^ activity was significantly reduced in corticosterone-treated mice. Unfortunately, the authors did not investigate the Ca^2+^ signaling of mPFC astrocytes in corticosterone-treated mice exposed to anxiogenic environments, such as the open arms of the EPM. Instead, they expressed the Gq-hM3Dq DREADD specifically in mPFC astrocytes to investigate the impact of boosting astrocyte Ca^2+^ signaling on behavior. Twenty minutes after i.p. application of the agonist CNO, they found a reduction in the anxiety elicited by corticosterone treatment, as well as an improvement in other mood behaviors impaired by corticosterone. In contrast, Gq activation of mPFC astrocytes in naive mice led to impaired performance, with increased anxiety levels, greater despair behavior and altered social interactions. These contrasting results are particularly interesting, as they illustrate how the same intervention on astrocytes can have different impacts on behavior, depending on the functional state of the neuron–astrocyte circuits.

The liver X receptor β (LXRβ) is a transcription factor expressed in various organs, including the brain, and deletion of this receptor has been shown to cause temporarily elevated levels of anxiety [63], although the cellular mechanism underlying this anxiogenic effect was unknown. Given that LXRβ is enriched in astrocytes cultures compared to neuronal cultures [64], a recent study used a GFAP promoter-mediated Cre-LoxP system to obtain conditional knockout mice of LXRβ in astrocytes and investigate the role of this astrocyte receptor in anxiety levels [65]. Deletion of LXRβ in astrocytes from adult male mice elicited an elevated anxious state when knockout mice were exposed to the OFT, EPM and LDB test. This anxiogenic state was accompanied by a possibly confounding reduced locomotor activity observed in the OFT (but see later in the article), and these mice did not exhibit depression like-behaviors when tested in a battery of behavioral tasks (forced swimming test, tail suspension test and sucrose preference test). Interestingly, the deficiency of LXRβ in adult female mice did not alter either anxious or depressive behaviors. These results were overall recapitulated, except for the decrease in locomotor activity, when the authors used an AAV strategy to delete astrocyte LXRβ locally in the mPFC. In layer V neurons of the mPFC from conditional knockout mice, the authors found an increase in the expression levels of the excitatory transmission marker vGlut1, without changes in the expression levels of the inhibitory transmission marker vGAT. Moreover, these neurons exhibited a notable elevation in the length and crossing of their basal dendrites, as well as an increase in excitatory transmission, as revealed by the elevated frequency of spontaneous excitatory post-synaptic currents (sEPSCs). This latter observation was also replicated in mice lacking LXRβ only in astrocytes from the mPFC. Finally, the authors found a decreased expression level of glutamate transporter-1 (GLT-1, also known as EAAT2) in mPFC from both types of knockout mice, an interesting observation as dysfunction of glutamatergic transmission is involved in various mood disorders, including anxiety [66]. In mice with local deletion of LXRβ in mPFC astrocytes, systemic administration of ceftriaxone, a β-lactam antibiotic that increases GLT-1 levels [67], was sufficient to restore normal levels of excitatory transmission and, more importantly, of anxiety-like behavior. While the recovery of excitatory transmission in mPFC brain slices was likely mediated by the rescue of GLT-1 expression in the mPFC, the systemic nature of the GLT-1 rescue tool used by the authors does not allow for an unambiguous conclusion that the effect on anxiety was mediated exclusively by the recovery of GLT-1 in the mPFC, as in other brain regions, basal levels of GLT-1 were likely increased. Indeed, in control mice treated with ceftriaxone, a partial anxiolytic effect was observed, but only in the OFT. In any case, these results suggest that deletion of LXRβ in mPFC astrocytes leads to several structural and functional changes in these cells, some of which may be directly involved in the control of anxiety levels.

GLT-1 and the glutamate–aspartate transporter 1 (GLAST-1, also known as EAAT1), mainly expressed in glial cells, are the predominant glutamate transporters in the brain [68]. Given that patients with major depression exhibit increased activity in the ventral anterior cingulate cortex, which is equivalent to the mouse IL cortex [53], in the study conducted by Fullana et al., the authors hypothesized that the downregulation of glutamate transporters could mimic the pathological hyperactivity in the IL [69]. Using microinjections of small interference RNAs (siRNAs) targeting GLAST-1 and GLT-1 in IL of male mice, the authors found a reduction of approximately 80% in the GLAST-1 and GLT-1 transporters, along with a slight reduction in the number of astrocytes. Notably, the reduced expression of GLAST-1 and GLT-1 in IL astrocytes led to depressive-like behavior without changes in anxiety levels. This lack of effect on anxiety-related behavior when the expression of GLAST-1 and GLT-1 was reduced in IL appears to be at odds with the data presented by Li et al. regarding the GLT-1 deficit in the mPFC of LXRβ knockout mice and the associated increase in anxiety manifestations [65]. The differences in the experimental conditions between the two studies (reduction in GLT-1 versus reduction in both GLAST-1 and GLT-1, a deficit in the entire mPFC versus a localized deficit in the IL cortex, and additional changes in gene expression associated with the LXRβ deficit, but not with the use of siRNAs for GLAST-1 and GLT-1) likely explain the apparently contrasting results observed. Indeed, the crucial role of glutamate transporters in regulating anxiety has been described in the PL cortex of male mice [70]. In this region of the PFC, local application of the glutamate transporter inhibitor TBOA elicited an increase in extracellular glutamate and the emergence of a higher anxious state in the OFT, while keeping the total distance traveled in this test unaltered. Although TBOA is a general inhibitor for all glutamate transporters, it is tempting to speculate that the effect of TBOA on anxiety is mediated by the inhibition of astrocyte glutamate transporters in the PL cortex. Overall, these results suggest that, in the different subregions of the PFC, impaired function of glutamate transporters, mainly expressed by astrocytes, impact anxiety-like behavior differently.

The ACC, located in the upper part of the medial prefrontal cortex, is a hub for emotional regulation, pain modulation and pain-induced anxiodepressive-related behaviors [71,72,73]. A recent study conducted by Wei et al. [74] investigated the role of ACC astrocytes in the comorbid expression of anxiety during the chronic inflammatory pain triggered in male rats by paw injection of complete Freund’s adjuvant (CFA). In the OFT and EPM test, CFA-treated rats displayed an elevated anxiety level compared to saline-injected rats. Notably, 30 min before the behavioral test, the activation of Gi-hM4Di DREADD receptors specifically expressed in ACC astrocytes slightly relieved the allodynia triggered by CFA and alleviated the anxious levels observed in the OFT and EPM test, without changing the total distance traveled in both tests [74]. Like in other studies summarized in this review, in the study by Wei et al., the Ca^2+^ signaling of astrocytes in response to Gi-DREADD activation was not investigated during the behavioral test, ruling out the possibility of correlating astrocyte Ca^2+^ signaling with the behavioral changes observed. Given that allodynia was also partially reduced in these experiments, the alleviated anxiety manifestations evoked by the modification of astrocyte Gi signaling might just reflect a contingent effect due to a decrease in the primary cause of elevated anxiety (in this case, the painful condition), and not a direct role in the regulation of pain-induced anxiety. PFC astrocyte regulation of pain-triggered anxiety remains an intriguing hypothesis, but further experiments are required to investigate it in more detail.

In mice that are conditionally knocked out for the astrocytic expression of the vesicular monoamine transporter 2 (aVMAT2 mice), a recent study showed that the absence of VMAT2 in astrocytes from the PFC favors the continuous degradation of dopamine (DA) by the monoamine oxidase B (MAOB), keeping cytosolic DA concentrations low, which drives the activity of the plasma membrane DA transporter OCT3 and, ultimately, decreases extracellular DA levels [75]. This hypo-DAergic condition was also associated with increased excitatory activity in the PFC, due to the loss of DA-mediated inhibition through the negative modulator D2 receptors. In a follow-up study [76], the authors found that astrocyte VMAT2 knockout mice showed anxiety-like behaviors in the OFT and EPM test, as well as an excessive grooming (a form of repetitive behavior typical in mouse models of obsessive–compulsive disorder (OCD) spectrum) even in their home cages. In the OFT, mice showed also a reduced locomotor activity, but only in the first 2 min of arena exploration, when anxiety levels are presumed to be highest. Interestingly, when VMAT2 was specifically re-expressed in PFC astrocytes using a lentiviral vector (an intervention that restored PFC DA levels and excitatory transmission [75]), both anxiety and grooming activity were rescued, suggesting that the impairment of PFC astrocytes is sufficient to regulate anxiety and grooming. Indeed, in VMAT2 floxed mice, the injection of a lentivial vector to specifically express Cre recombinase in PFC astrocytes, leading to the deletion of VMAT2 only in these astrocytes, recapitulated all the findings in aVMAT2 mice, from reduced DA levels and increased neuronal activity in the PFC to pathological grooming and anxiety-like behaviors. Overall, these results suggest that astrocytes from the mPFC participate in the control of anxiety levels by regulating extracellular levels of not only glutamate but also DA.

## 4. Amygdala

The amygdala, located anterior to the hippocampus in the medial temporal lobe, is a brain region associated with emotional processing of negative stimuli (e.g., fear, anxiety) [77,78,79,80]. The amygdala is a complex anatomical and functional region composed of the basolateral amygdala (BLA) and the central amygdala (CeA), with the latter subdivided into the lateral and medial central amygdala (CeL and CeM amygdala). The BLA is mainly a glutamatergic nucleus involved in the mechanisms of fear associative processing while the CeA, which is mainly a GABAergic region, constitutes the primary output region that project from CeM to outside the amygdala to regulate the fear response [77]. Within the amygdala, different neuronal types can modulate anxiety-related behavior in various ways [81,82]. Like neurons, astrocytes in the amygdala can also influence anxiety levels. In a study conducted by Xiao et al. [83], the authors used the unpredictable chronic mild stress (UCMS) model to induce an elevated level of anxiety in male mice. The experimental design of the UCMS model includes a variety of protocols, with the common feature being the randomized exposure of mice to mild stressors over a three- to four-week period. This experimental model has been classically used to elicit depressive behavior in mice [84,85]. In the study performed by Xiao et al., mice were stressed by exposure to physical restraint, wet environments, squeezing social interactions and light during their natural dark phase. Under these conditions, mice displayed elevated levels of anxiety in the OFT and EPM tests [83], as well as an increase in serum corticosterone levels. In stressed mice, acute optogenetic stimulation of BLA astrocytes expressing ChR2, lasting five or three minutes during the performances in the OFT and EPM test, respectively, slightly attenuated anxiety levels only in the OFT, with no differences observed in the distance traveled in this test. In contrast, when the presentation of the stressors was accompanied by optogenetic stimulation of BLA astrocytes for 10 min every two days during the three weeks of the chronic stress protocol, the chronic stimulation of astrocytes exerted an anxiolytic effect in both OFT and EPM tests, without changes in total locomotor activity in the OFT, and restored blood levels of corticosterone [83]. The difference in the strength of the effects obtained with acute and chronic optogenetic stimulation of astrocytes suggests that the most significant relief of the anxiogenic state in stressed mice likely relies on changes in the gene expression of activated astrocytes, elicited by the repetitive stimulation of Ca^2+^ and/or other intracellular signaling pathways related to ChR2 activation in astrocytes. A putative candidate to explore was the expression level of the glutamate transporter GLT-1. In a recent paper, the same authors further investigated the role of BLA astrocytes in chronic stress-mediated anxiety levels [86] and found that, after 4 weeks of the UCMS protocol, stressed male mice showed an upregulation of GLT-1 in astrocytes from the BLA, with no changes in astrocyte GLT-1 expression in PFC and hippocampus. Likewise, compared to naive mice, stressed mice showed higher Ca^2+^ activity in glutamatergic neurons from the BLA, both in freely moving conditions and during 10 s tail suspension. The use of pharmacological or genetic tools (overexpression) to boost astrocyte GLT-1 activity in BLA was sufficient to induce anxiety-like behavior in naive mice in the EPM test and OFT, without changing the distance traveled in the OFT. Conversely, reducing astrocyte GLT-1 activity with pharmacological or genetic tools (AAV-delivered shRNA) restored stress-mediated anxiety levels to those observed in naive mice, while keeping the distance traveled in the OFT unaltered, and reversed the hyperactivity of glutamatergic neurons in the BLA triggered by chronic stress. Given that the upregulation of GLT-1 activity decreases extracellular glutamate levels, the authors advanced the hypothesis that the hyperactivity of glutamatergic neurons was likely mediated by a decrease in local GABAergic neuron activity. Additional experiments are needed to corroborate this hypothesis. A recent paper further investigated the contribution of BLA astrocytes to anxiety and fear memory, using a Gq-based chemogenetic approach to activate astrocytes [87]. In this study, chemogenetic activation of BLA astrocytes 30 min before the behavioral tests contributed to the formation, but not the recall, of auditory cued fear memory without significant effects on anxiety in male mice [87]. This result aligns with the slight effect observed by Xiao et al. in anxiety behavior after acute optogenetic activation of BLA astrocytes. Although the experimental conditions in these two studies are strikingly different (basal anxiety versus stress-triggered anxiety, chemogenetic versus optogenetic astrocyte activation), it is tempting to speculate that only chronic manipulations of BLA astrocytes are efficient enough to modulate anxiety-like behavior in mice. Whether chronic chemogenetic manipulation of BLA astrocytes affects anxiety awaits further studies.

Astrocytes in the CeL respond with somatic Ca^2+^ elevations to oxytocin (OXT) [88], a neuropeptide widely recognized to play crucial roles in social and emotional behaviors, including anxiety disorders [89]. In brain slices containing the CeL from female and male mice, the astrocyte Ca^2+^ response to oxytocin occurs through the activation of the OXT receptor (OXTR), a GPCR typically associated with Gq signaling, and the cooperation of gap junctions to propagate the Ca^2+^ signaling in the astrocyte network [88]. The functional consequences of this astrocyte activation are an increased activity of GABAergic CeL neurons and an ensuing increase in the inhibitory tone on CeM neurons, all of these effects being impaired by the deletion of the OXTR specifically in CeL astrocytes. To investigate the behavioral consequences of the astrocyte OXTR signaling, the authors then used a model of neuropathic pain elicited by spared nerve injury surgery (SNI) that is associated with mechanical pain hypersensitivity (allodynia) and elevated levels of anxiety. Interestingly, the local application of an OXTR agonist in the CeA of sham-operated and SNI mice did not change the sensitivity to mechanical pain in these mice, but it relieved the elevated anxiety state displayed by SNI mice in the EPM test, without changing the level of anxiety in sham-operated mice. Moreover, in mice lacking OXTR specifically in CeL astrocytes, the anxiolytic effect exerted by the OXTR activation in SNI mice was completely lost. Most importantly, the deletion of OXTR in CeL astrocytes from sham-operated mice was sufficient to trigger an elevated level of basal anxiety compared to sham-operated mice expressing OXTR in astrocytes. All the changes observed in anxiety levels occurred without changes in the total distance traveled in the EPM test. Overall, these results illustrate the anxiolytic effects of oxytocin signaling in CeL astrocytes under basal conditions and during painful conditions [88].

Within the CeM, the Gq-chemogenetic activation of astrocytes 30 min before behavioral tests has been shown to reduce fear expression after a fear conditioning paradigm without changing anxiety levels of male mice in the EPM test [90]. This finding aligns with the lack of effect of acute chemogenetic activation of BLA astrocytes on anxiety levels [87]. We have previously seen that the BLA is activated by stressful conditions associated with the release of corticosterone and activation of the GR. Within the amygdala, the CeA is also associated with stress and anxiety-related pathologies [91]. A recent study by Wiktorowska et al. reported that the use of a lentiviral vector to express in CeA astrocytes an shRNA specific for the GR reduced, by approximately 70%, the number of CeA astrocytes expressing GR and, in those astrocytes that still presented detectable GR expression, the levels were reduced by approximately three-fold [92]. In these mice, the acquisition of fear conditioning was similar to that in shRNA control-expressing mice. However, the recall of this fear was reduced and, similarly, basal anxiety levels were attenuated, as revealed by the significant increase in the time spent in the center of the open field and the tendency to spend more time in the light box of the LDB test and the open arms of the EPM test. These changes in anxiety levels occurred without significant changes in locomotor activity in the OFT. As in the BLA, the involvement of GLT-1 in anxiety has been suggested in the CeA. However, in contrast to that described previously in BLA, the local pharmacological blockade of GLT-1 with its inhibitor DHK within the CeA resulted in boosted anxiety-like symptoms in male rats [93]. Overall, these studies suggest that astrocytes in the amygdala actively participate in the regulation and control of anxiety levels through various complex mechanisms.

## 5. Striatum

Anxiety can manifest in obsessive–compulsive disorders (OCD), a psychiatric disorder that appears to be associated with cortico–fronto–striatal circuits and their limbic connections [94]. Knockout female and male mice for SAPAP3, a cytosolic scaffold protein highly expressed in both neurons and astrocytes from the striatum and associated with obsessive–compulsive disorders (OCD) in humans and repetitive behavior, show OCD-like phenotypes (i.e., abnormal, repetitive self-grooming) and anxiety-like behavior [95]. These phenotypes are accompanied by a reduction in locomotor activity in the OFT. In these mice, the re-expression of SAPAP3 in striatal neurons or astrocytes rescued the OCD-like behavior but anxiety levels were ameliorated only after re-expression of SAPAP3 in neurons, not in astrocytes [95], suggesting that the higher anxiety of SAPAP3 knockout mice is astrocyte-independent. In striatal astrocytes, SAPAP3 interacts with proteins related with actin cytoskeleton and Gi proteins, and the absence of SAPAP3 in knockout mice leads to impaired astrocyte morphology and territory occupied due to a reduction in the intensity of the actin cytoskeleton at the periphery of the cell. Re-expression of SAPAP3 in striatal astrocytes rescued the astrocyte morphology. Although the increase in anxiety in SAPAP3 knockout mice seems to be astrocyte-independent, a subsequent study by the same research group found that striatal astrocyte manipulations could relieve anxiety-like behaviors in SAPAP3 knockout mice [96]. In an attempt to rescue astrocyte morphology, the authors used chronic stimulation of the Gi pathway in astrocytes. To achieve this, they employed an AAV strategy to express the Gi-hM4Dq DREADD receptor in striatal astrocytes and administered the DREADD agonist deschloroclozapine (DCZ) every three days for approximately four months. The Gi striatal astrocyte manipulation resulted in rescued astrocyte morphology, with a partial recovery of the astrocyte territory. Interestingly, chronic activation of the Gi pathway in SAPAP3 knockout mice not only recovered astrocyte morphology, but also alleviated both OCD and anxiety phenotypes, recovering also locomotor activity. In contrast, a similar manipulation of the Gq pathway in SAPAP3 knockout mice did not rescue either the behaviors or astrocyte morphology. The behavioral changes induced by chronic Gi stimulation were accompanied by a reduction in neuronal hyperactivity, as measured by the marker of sustained neuronal activity ΔFosB and the number of action potentials evoked by depolarizing current injections, observed in both the striatum and cortex of SAPAP3 knockout mice. These studies illustrate how astrocytes can be exploited for novel therapeutic strategies to treat neuropsychiatric disorders that, at first, seem unrelated to astrocyte function.

## 6. Hypothalamus

Another brain structure associated with the regulation of anxiety levels, due at least in part to the neural production of neuroactive molecules involved in anxiety control, such as OXT and CRF, is the hypothalamus [97]. Like other brain structures, the hypothalamus is a complex anatomical and functional region in which different nuclei, such as the ventromedial hypothalamus (VMH) and paraventricular nucleus (PVN), among others, can be identified [98]. A recent study exploring the hypothalamus investigated the association between chronic stress and anxiety [99]. Using a UCMS model, in which male mice were exposed for 8 weeks to squeezing social interactions and wet and high position environments, the authors found that the elevated anxiety levels triggered by chronic stress in this model were due to increased inhibitory pressure from the bed nucleus of the stria terminalis (BNST) onto the steroidogenic factor-1 (SF-1) positive neurons of the VMH, in terms of increased VMH extracellular GABA levels and the frequency of spontaneous IPSCs in SF-1 neurons. Interestingly, in this model of chronic stress, astrocytes from the VMH exhibited increased expression of the cellular activation marker c-Fos. In a follow-up study, the impact of VMH astrocytes in the chronic stress-induced anxiety was investigated [100]. The authors used an AAV strategy to specifically express Gi and Gq-mediated DREADDs in VMH astrocytes. In these mice, the activation of the Gi pathway in VMH astrocytes alleviated stress-induced anxiety levels, both in the OFT and EPM tests. Unfortunately, the temporal details of the Gi activation (i.e., the schedule of the i.p. CNO injections during behavioral testing) were not clearly sated by the authors. Interestingly, in the absence of chronic stressors, the activation of the Gq pathway in VMH astrocytes was sufficient to induce significant anxiety-like behavior in both the OFT and EPM tests. Although the total distance traveled was not assessed after astrocyte manipulation, these results suggest that VMH astrocytes can bidirectionally regulate anxiety levels. Further experiments are necessary to better characterize the mechanisms through which VMH astrocytes modulate anxiety states.

Astrocytes from the PVN of the hypothalamus have been shown to participate in the mechanism of anxiety triggered by pro-inflammatory cytokines in rats [101]. Specifically, 7 days after intracerebroventricular (i.c.v.) infusion of IL-1β, male rats exhibited elevated anxiety levels in the OFT and EPM test, along with an increased expression of GFAP and elevated GABA content in PVN astrocytes. Similarly, microdialysis analysis revealed an increase in the GABA levels within the PVN. Previous local bilateral intra-PVN infusion of _L_-α-aminoadipate (_L_-AAA), a gliotoxin that interferes with astrocyte function [102], significantly ameliorated inflammation-mediated anxiety-like behavior in the EPM test, while relief in the OFT, although observed to some extent, did not reach statistical significance. Although general locomotor activity was not assessed in these experiments, these results suggest that PVN astrocytes contribute to the anxiety associated with brain inflammation. The gliotoxin pretreatment also reduced the PVN astrocyte GFAP and GABA contents elevated by IL-1β, as well as the extracellular GABA levels. To test whether activated PVN astrocytes could modulate anxiety behavior by releasing GABA through Best-1 channels, a type of anion channel known to mediate GABA and glutamate release from astrocytes [103], the authors investigated the effects of the Best-1 blocker NPPB in IL-1β-induced anxiety. Local infusion of NPPB within the PVN significantly ameliorated anxiety-like behavior and reduced GABA levels within the PVN, supporting the hypothesis of the participation of astrocyte GABA release in the inflammatory cytokine-induced anxiety-like behavior.

I previously reviewed how astrocytes play a relevant role in the anxiolytic effects of OXT in the CeA. A recent study by Meinung et al. provided further evidence of the central role of astrocytes in the anxiolytic effects exerted by OXT in the PVN [104]. As expected [105], local application of OXT within the PVN of male rats exerted anxiolytic actions when the rats were tested in the EPM and LDB tests, with no changes in total locomotor activity in both tests. Interestingly, when the authors used an AAV strategy to target the expression of shRNA for OXTRs to astrocytes, the reduced expression of OXTRs in PVN astrocytes abolished the anxiolytic effects of OXT. A similar effect was observed when the authors downregulated the protein Gem, located downstream of OXTR, which is crucial for the regulation of cytoskeletal morphology of OXT-treated astrocytes. These findings illustrate how the well-described anxiolytic properties of OXT within the PVN are mediated by PVN astrocytes.

Recently, the role of PVN astrocytes in chronic pancreatitis (CP)-mediated pain and anxiety has been described by Luo et al. [106]. This pain model, in which CP was induced by i.p. injection of caerulein in male mice, is associated with pancreas atrophy, high levels of inflammation, abdominal pain hypersensitivity (revealed by the von Frey test) and increased anxiety levels (measured by the OFT and EPM test). Given that, in models of visceral pain, the PVN seems to play a pivotal role in the control of abdominal allodynia, the authors investigated the involvement of PVN astrocytes in caerulein-triggered pain and anxiety. In mice with CP, the authors found significant astrogliosis, indicated by increased expression of GFAP in the PVN of these mice, reduced expression of GLT-1 and aberrant spontaneous Ca^2+^ signaling in astrocytes of the PVN, as detected by the AAV-mediated expression of GCaMP6s and in vivo fiber photometry analysis. Interestingly, low abdominal stimulation significantly increased PVN astrocyte Ca^2+^ signaling in CP mice, but not in sham mice. To evaluate the effect of reducing astrocyte Ca^2+^ signaling on abdominal allodynia and anxiety-like behavior, the authors used an unconventional Gi-DREADD receptor, the kappa-opioid receptor (KORD), activated by the pharmacologically inert ligand Salvinorin B [107], which reduced basal Ca^2+^ levels and spontaneous Ca^2+^ elevations in brain slices. In CP mice expressing KORD in PVN astrocytes and treated with Salvinorin B, both abdominal pain hypersensitivity and anxiety were alleviated, although controls for assessing unaltered locomotor activity are lacking in this study. In these mice, the hyperactivity of glutamatergic neurons in the PVN, triggered by caerulein, was also corrected, along with the reduced expression of the astrocyte glutamate transporter GLT-1. In brain slices, the authors found that inhibition of the glutamate transporter was sufficient to increase the excitability of glutamatergic PVN neurons. Overall, these results suggest that reducing astrocyte Ca^2+^ signaling in CP mice reduces pain and anxiety by rescuing GLT-1 expression levels and correcting neuronal hyperexcitability. In support of this hypothesis, the authors found that the systemic treatment of CP mice with LDN-212320, an agonist of GLT-1, alleviated the hyperactivity of glutamatergic neurons in the PVN, as well as the abdominal pain and anxiety-like behavior in these mice. As seen in the PFC, astrocyte-related modifications affect both allodynia and pain-mediated anxiety, raising the possibility of an indirect effect on anxiety when PVN astrocytes are inhibited. However, given that PVN astrocytes can modulate anxiety not triggered by pain, the possibility remains that PVN astrocytes directly regulate CP-triggered anxiety.

## 7. Lateral Septum

The lateral septum (LS), a forebrain region mainly composed of GABAergic neurons, is critical in stress responses and anxiety, having bidirectional control of anxiety and depression [108]. A recent investigation explored the impact of LS astrocyte activity during stress responses and anxiety in male mice [109]; however, it is important to note that the potential confounding effects on locomotion were not assessed during the manipulation of LS astrocyte activity. In this study, the authors used an AAV-based strategy to express the plasma membrane Ca^2+^ pump hPMCA2w/b specifically in LS astrocytes. This pump has been shown to reduce Ca^2+^ signaling in astrocytes and interfere with astrocyte functions in various cerebral circuits [110,111,112,113,114,115,116,117,118,119,120,121]. In naive mice, reducing Ca^2+^ signaling in LS astrocytes did not affect basal anxiety levels observed in the OFT and LDB test, and nor did it alter social interactions. However, in mice that were acutely exposed to aggressive male mice, the expression of hPMCA2w/b in astrocytes located in the more dorsal parts of the LS alleviated the increased anxiety state induced by the acute social defeat stress (ASDS) paradigm, at least in the LDB test. Similarly, the social avoidance associated with ASDS was reduced when Ca^2+^ signal in dorsal LS astrocytes was limited, increasing the proportion of mice that were resistant to social defeat stress. Interestingly, when social stress was chronic, expression of the Ca^2+^ pump in dorsal LS astrocytes only partially ameliorated social avoidance, with no changes in the percentages of susceptible and resilient mice, and it did not recover anxiety levels. Modifying astrocytes in the ventral part of the LS had no effect on stress-induced behaviors. These results suggest that limiting Ca^2+^ signaling in dorsal LS astrocytes has an anxiolytic effect in temporally limited paradigms of social defeat stress. Consistent with these findings, Gq-DREADD chemogenetic activation of dorsal, but not ventral, LS astrocytes induced an anxiogenic effect. Specifically, in naive mice, Gq-DREADD activation did not change anxiety-like behavior or social interactions. However, when mice were subjected to subthreshold social defeat stress (SSDS), a condition that alone did not significantly alter mouse behavior, Gq-DREADD activation of dorsal LS astrocytes enhanced anxiety-like behavior in the OFT and EPM test, as well as social avoidance, resulting in a significant increase in the percentage of mice susceptible to social defeat stress. In accordance with these findings, using in vivo fiber photometry in mice expressing GCaMP6f in LS astrocytes, the authors found that dorsal, ventral and intermediate LS astrocytes selectively responded with Ca^2+^ elevations to various dangerous, frightening and stress-related stimuli, including attacks by male aggressors in social defeat stress paradigms. In brain slices, chemogenetic activation of dorsal or intermediate LS astrocytes suppressed excitatory transmission in neighboring neurons via adenosine A1R activation, while inhibitory transmission was increased only with chemogenetic activation of intermediate LS astrocytes via adenosine A2AR activation. In contrast, chemogenetic activation of dorsal, but not intermediate, LS astrocytes facilitated the activation of distant neurons within the same or adjacent regions. Unfortunately, the study did not assess whether these effects observed in brain slices contribute to the regulation of in vivo anxiety levels. However, it appears that the dynamic interactions between astrocytes and neurons within the LS are crucial for regulating stress responses, including anxiety behavior.

## 8. Lateral Habenula

The lateral habenula (LHb) is a subregion of the small habenula nucleus. Located above the thalamus, the habenula is a medial brain region crucial for value-guided behavior [122]. One of the most widely recognized functions of LHb activity is its encoding of aversive, negative emotional stimuli, and it plays an active role in aversive behaviors and depression [123]. A recent study investigated the involvement of LHb astrocytes in the modulation of anxiety-like behavior in male mice [124]. Exploiting the natural fear of mice toward glass marbles, the authors created a non-classical, single-compartment anxiogenic environment, the all-marble cage (AMC), in which the classical bedding at the floor of the cage is substituted by glass marbles. In this high-anxiogenic environment, recordings of local field potentials from the LHb showed an increase in theta band activity, which was also observed during the spontaneous entries into the anxiogenic center zone of the classical OFT. Using fiber photometry in mice expressing FRET-based sensors to measure Ca^2+^ or pH in astrocytes in vivo, the authors then investigated cytosolic changes in astrocytes in anxiogenic AMC. Given that changes in fluorescence from certain FRET fluorophores can arise due to variations in pH and brain blood volume, the authors made a series of adjustments to their measurements. Following these adjustments, they concluded that, upon placing the mouse in the anxiogenic AMC environment, LHb astrocytes responded with an immediate Ca^2+^ increase, followed by a sustained decrease in basal Ca^2+^ levels and with an acidification of their cytosol. Unfortunately, the mechanism underlying this unexpected Ca^2+^ decrease was not studied by the authors, and nor was the potential effect of this decrease on anxious behavior. Instead, the acidification of astrocytes observed in the anxiogenic environment was hypothesized to be an active component of the anxiogenic mechanism. Photoactivation of the proton pump ArchT, specifically in LHb astrocytes, counteracted the acidification evoked by the glass marbles and, interestingly, markedly decreased theta band activity in the LHb. Considering that astrocyte acidification in other brain regions favors gliotransmission and has a remarkable effect on astrocyte-to-neuron communication [125], the results obtained with the proton pump suggest that gliotransmission in the LHb could contribute to the elevated anxiety induced by glass marbles. Although additional experiments are needed to better understand the role of LHb astrocytes in anxious behavior, the study by Tan et al. [124] directs our attention to astrocytes in this brain region and their role in encoding emotional states during anxiety.

Other pathological conditions comorbid with depression and anxiety include alcohol use disorders [126]. Accordingly, in the study conducted by Kang et al. [127], an exacerbation of depression and anxiety-like behavior (the latter measured in the EPM test) was observed in alcohol-withdrawn rats. Interestingly, compared to control rats, alcohol-withdrawn rats exhibited significantly lower astrocyte expression of GLT-1 in the LHb and, consistently, the decay time of neuronal spontaneous excitatory postsynaptic currents was increased in the LHb of rats treated with ethanol [127]. The i.p. ceftriaxone treatment in alcohol-withdrawn rats significantly restored GLT-1 expression in the LHb and normalized the hyperexcitability of LHb neurons. Notably, the heightened depression and anxiety-like behavior associated with alcohol withdrawal was also prevented by ceftriaxone treatment, without changing the total locomotor activity in the EPM test. Although we cannot rule out the possibility of off-target effects outside LHb during the systemic administration of ceftriaxone, these findings suggest that the reduced expression of the astrocyte glutamate transporter GLT-1 in the LHb may play a prominent role in the psychiatric disorders comorbid with ethanol withdrawal.

## 9. Periaqueductal Gray

The periaqueducal gray (PAG) is a region located in the midbrain that participates in relevant survival behaviors, such as defensive and aggressive behaviors, as well as in the regulation of mood-related disorders, like anxiety and depression [128]. A recent study by Yan et al. investigated the role of PAG astrocytes in anxiety-like behavior [129]; however, as in other studies, the potential confounding effects on general locomotion were not assessed during the manipulation of PAG astrocyte activity. The authors first demonstrated that a stressful foot shock protocol triggered, as expected, an anxiety-like state in male mice, as revealed in the LDB, OFT and EPM tests. This acute stress also influenced the activity of the ventrolateral PAG (vlPAG) serotoninergic (5-HTergic) neurons in various ways. In particular, exposure to foot shock acute stress boosted the intrinsic excitability of vlPAG neurons in brain slices, leading to a higher action potential firing frequency in response to a constant injected current. It also favored excitatory transmission in these cells by increasing the frequency and amplitude of miniature excitatory post-synaptic currents (mEPSCs), and enhanced their excitability by increasing the area under the curve of the so-called slow inward currents (SICs) evoked in the absence of extracellular Ca^2+^ [130]. SICs are excitatory currents caused by glutamate acting on extrasynaptic glutamate NMDA receptors, and occur through the activation of astrocytes via Ca^2+^-dependent and Ca^2+^-independent mechanisms [131,132,133]. Since astrocyte activation can also modify synaptic transmission [134,135], this raised the question of whether astrocytes participate in the changes evoked in vlPAG neurons by acute stress and in the regulation of anxiety levels. To investigate whether astrocyte activation in the vlPAG favors anxiety-like states, the authors used a chemogenetic Gq-hM3Dq DREADD-dependent protocol to activate vlPAG astrocytes. Gq signaling activation in vlPAG astrocytes induced a heightened anxious state in the OFT and LDB test compared to control mice. Interestingly, similarly to the effects observed after the stressful foot shock protocol, the Gq activation of vlPAG astrocytes per se also evoked an increase in intrinsic excitability, excitatory transmission (in terms of frequency of mEPSCs) and the amplitude of SICs in vlPAG 5-HTergic neurons. These results suggest that Gq activation of vlPAG astrocytes exerts anxiogenic effects, likely by increasing the activity of vlPAG 5-HTergic neurons, as occurs during acute stress. The authors then used the Gi-hM4Di DREADD activation in astrocytes to “inhibit” them. In these experiments, the activation of the Gi pathway in vlPAG astrocytes 30 min before the acute stress protocol ameliorated the anxiogenic state triggered by stress, at least in the LDB and EPM tests. Similarly, Gi activation of astrocytes attenuated the increases in synaptic transmission and excitability of 5-HTergic vlPAG neurons. Unfortunately, in this study as well, the in vivo Ca^2+^ signaling in vlPAG astrocytes after Gq or Gi activation was not investigated, and nor were the Ca^2+^ signals in astrocytes when mice were exposed to foot shock stress in the absence of prior manipulations of astrocyte signaling pathways. Although we cannot deduce the role of astrocyte Ca^2+^ signaling in these Gq- and Gi-activation experiments, these results overall suggest that the manipulation of astrocytes in the vlPAG can bidirectionally influence basal and stress-evoked anxiety levels.

## 10. Zona Incerta

An upstream region of the vPAG is the subthalamic zona incerta (ZI). The ZI is a multisensory convergence area that functions as an integrative node for the global modulation of behaviors and physiological states, including anxiety levels [136,137,138]. A recent study conducted in both female and male mice with a laparotomy surgical intervention investigated the role of ZI astrocytes in surgery-induced anxiety [139]. One day after this type of abdominal surgery, mice exhibited an elevated anxiety state in the OFT and EPM test. When the authors examined astrocytes in these mice, they observed an increased reactive state of astrocytes in the ZI, as well as abnormal Ca^2+^ activity of these astrocytes in in vivo fiber photometry experiments. Compared to sham mice, surgery-treated mice exhibited an increased frequency of Ca^2+^ elevations in ZI astrocytes, although with lower amplitude values. The ZI is a largely inhibitory region, and the activity of GABAergic neurons in the ZI was reduced after abdominal surgery, as indicated by decreased c-Fos expression and in vivo Ca^2+^ activity. Similarly, with the use of microelectrode arrays, authors found reduced in vivo neuronal firing rates. Interestingly, when ZI astrocyte Ca^2+^ activity was limited by the exogenous expression of the plasma membrane Ca^2+^ pump hPMCA2w/b, basal anxiety levels remained unaffected, while postoperative anxiety was alleviated without significant changes in the total distance traveled in the OFT. Similarly, the reduced Ca^2+^ activity of GABAergic neurons was rescued by the expression of the Ca^2+^ pump in astrocytes, suggesting that surgery-induced dysregulated Ca^2+^ activity in astrocytes leads to the deactivation of GABAergic neurons in the ZI. Then, the authors provided evidence that this inhibitory action on GABAergic neurons after surgery occurs through the downregulation of the expression of the astrocyte GABA transporter GAT3, the subsequent increase in extracellular GABA levels, and the ensuing augmented inhibitory tonic current in GABAergic neurons. In support of this hypothesis, the authors found that exogenously forced expression of GAT3 in ZI astrocytes of surgery-treated mice ameliorated postoperative anxiety, as did chemogenetic activation of ZI GABAergic neurons or optogenetic activation of ZI GABAergic projections to the medial raphe nucleus but not to the vPAG. Overall, these results illustrate how ZI astrocytes actively participate in the anxiety triggered by postoperative surgery.

## 11. Ventral Tegmental Area

Dysfunction of the PI3K-AKT/mTOR signaling has been associated with various psychiatric disorders, including depression and anxiety, and the antidepressant actions of some fast-acting and long-lasting antidepressants, such as ketamine, require the activation of the mTOR pathway [140]. mTOR is a serine/threonine kinase present in two mTOR complexes, mTORC1 and mTORC2. The obligatory scaffolding proteins Raptor and Rictor are components of the mTORC1 and mTORC2 complexes, respectively. While most studies have focused on mTOR in neurons, the role of mTOR in astrocytes has only recently been investigated. Using AAV-mediated CRE expression in astrocytes from Raptor^flox/flox^ or Rictor^flox/flox^ mice, a recent study deleted Raptor or Rictor in astrocytes from the ventral tegmental area (VTA), thereby disrupting mTORC1 and mTORC2 complexes, respectively [141]. The VTA is a midbrain region crucial for the reward system and is also involved in mood disorders, including anxiety [142]. Selective disruption of mTORC1 or mTORC2 in VTA astrocytes increased overall locomotor activity, without significantly affecting anxiety-like behavior in the OFT. In contrast, in the EPM test, Rictor (mTORC2) knockout mice showed a slight anxiolytic effect. Although the time spent in the open arms and the number of entries into these arms did not change, the total number of entries, a parameter associated with anxiety and reversed by treatment with anxiolytics, was reduced in these mice. Overall, these results suggest a minor but present contribution of VTA astrocyte mTORC2 to anxiogenic levels.

In the VTA, AAV-CRE-mediated deletion of GLT-1 in astrocytes also affected anxiety levels [143]. In these mice, deletion of astrocyte GLT-1 led to a marked increase in exploration of the center of the open field, with enhanced locomotor activity that did not reach statistical significance. These results suggest the participation of VTA astrocytes in the regulation of avoidance behaviors.

## 12. Genetic Alterations

In this section, I will summarize the results obtained from studies investigating anxiety levels in mice knockout for different astrocyte proteins (Figure 3). A recent study conducted by Jia et al. in GLT-1 knockout mice in astrocytes, generated by crossing GLT-1^flox/flox^ mice with GFAP^Cre^ mice, found that astrocyte GLT-1 deficiency leads to altered emotional regulation of depression, fear expression and anxiety [144]. Regarding anxiety, these mice displayed a decreased anxious state in the EPM test, with unaltered locomotor activity measured in the OFT. The authors used immunohistochemical analysis to confirm reduced levels of GLT-1 in three brain regions which are important for emotional regulation (PFC, striatum and hippocampus). These results are in line with the anxiolytic effect of GLT-1 deletion in VTA astrocytes [143], but they appear to contrast with the anxiogenic effects locally induced by the pharmacological blockade of GLT-1 in the mPFC [70] or central amygdala. The opposing results may be explained if the anxiolytic effect of GLT-1 deficiency in astrocytes across the whole brain largely compensates for the anxiogenic outcome of reduced GLT-1 activity specifically in diverse brain circuits. The results presented by Jia et al. also seem to contrast with a previous study involving conditional knockout mice for GLT-1 in astrocytes, which revealed no statistically significant changes in the anxiety-related behavior of these mice [145]. However, these mice showed a tendency to spend more time in the center of the OFT and in the open arms of the EPM test [145]. Differences in the extent of genetic deletion achieved by the conditional and non-conditional knockout strategies could explain the contrasting findings observed in these two studies.

A recent study by Lu et al. investigated the role of deleting astrocyte GRs in stress vulnerability, depression and anxiety manifestation [61]. To generate conditional knockout mice for the GR gene specifically in astrocytes, authors crossed GR^flox/flox^ mice with Fgfr3^Cre^ mice [146] (see later for further discussion). In these knockout mice, they observed manifestations of depressive behavior under basal conditions and after social stress. Moreover, an increased basal anxiety state was noted in these mice, as indicated by reduced time spent in the open arms of the EPM and increased time spent in the dark compartment of the LDB test. There were no changes in the time spent in the center of the OFT, but the mice exhibited reduced overall locomotion in this test. Given that corticosterone, the agonist of GR, is the main stress hormone, these results suggest that astrocyte GRs participate in the coping responses of animals to buffer the consequences of stress. When the authors used an AAV strategy to limit the deletion of GR to astrocytes in the mPFC, the mice displayed depressive-like manifestations without changes in anxiety levels. This observation suggests that astrocyte GR expression in regions other than the mPFC may serve as a coping mechanism to regulate anxiety levels. However, the concept of an anxiolytic effect associated with the activity of astrocyte GRs is incomplete, with the reduction in astrocyte GRs in CeA showing an anxiolytic effect on fear-mediated anxiety [92], as I have already discussed. Unfortunately, in the study by Lu et al., the authors did not further investigate the mechanisms underlying anxiety regulation. Instead, they focused on depressive mechanisms and provided evidence that stress evokes Ca^2+^ responses in mPFC astrocytes through the activation of GR, leading to the activation of the IP3K/Akt signaling pathway and the release of ATP from lysosomes, which ultimately contributes to the attenuation of depressive-like behavior. These stress-mediated Ca^2+^ responses in mPFC astrocytes are in line with the abnormal increase in Ca^2+^ signaling observed in mPFC astrocytes from corticosterone-treated mice exploring an open field [62]. Interestingly, the same authors used a similar genetic approach with Fgfr3^Cre^ mice to delete the RNA demethylase ALKBH5 [147], an enzyme that counteracts the N6-methyladenosine (m6A) methylation of RNAs that regulate gene expression [148]. The deletion of ALKBH5 resulted in a decrease in depression-related behavior under basal conditions and after social stress, and an increase in the basal anxiety levels. This increase was, however, slight, as only the time spent in the open arms of the EPM test was increased without changes in the time spent in the light box of the LDB test. The deletion of ALKBH5 also resulted in a slight reduction in general locomotion in the OFT. When ALKBH5 was deleted specifically in mPFC astrocytes, antidepressant-like behavior was still observed, but the anxiety levels of the knockout mice remained unchanged in both the EPM and LDB tests. Notably, in this study, the authors mentioned that Fgfr3, the protein whose promoter was used to target Cre expression to astrocytes, is also expressed in the interneurons of the olfactory bulb. To validate their results, authors deleted ALKBH5 in neurons of the olfactory bulb and found that in these mice, depression-related behavior remained unchanged, while anxiety levels increased. Although the deletion of ALKBH5 was not restricted to interneurons of the olfactory bulb in these experiments, these results suggest that the observed effects on depressive behavior were likely mediated by ALKBH5 deletion in astrocytes. However, some concerns remain regarding the role that astrocyte ALKBH5 plays in anxiety behavior, as the deletion of ALKBH5 in neurons of the olfactory bulb similarly resulted in an increased anxiety state. Whether the deletion of GR in interneurons of the olfactory bulb exerts an anxiogenic effect has not been tested and additional experiments are necessary to confirm the results obtained in mice expressing Cre under control of the Fgfr3 gene [61].

Deletions of two other genes specifically in astrocytes also affected anxiety levels in knockout mice. The deletion of the *Mtnr1b* gene, which encodes the melatonin receptor type 2 (MTNR1B), increased anxiety expression when knockout mice were tested in the OFT and EPM [149]. Although these results are consistent with the anxiolytic actions of the MTNR1B agonist UCM765 [150], the compromised locomotor activity displayed by these mice in the OFT raises concerns about the role of astrocytic MTR1B receptors in regulating anxiety states. Additional experiments are needed to investigate the impact of astrocytes in the modulation of anxiety by melatonin. An elevated anxiety state was also observed after the deletion of the gene encoding the vesicular nucleotide transporter Vnut, as indicated by the reduced number of entries in the center zone of the OFT, accompanied by an unaltered total distance traveled in this test, in conditional knockout Vnut mice [151]. Vnut in astrocytes mediates the loading of ATP into secretory vesicles, and ATP has been shown to play a key role in mood disorders such as depression [152]. Interestingly, the anxiogenic effect of Vnut deletion was observed in female, but not male, mice. In contrast, Vnut deficiency boosted depressive-like behavior without apparent sex differences. When the authors used an AAV-based strategy to restrict Vnut deletion to astrocytes in the nucleus accumbens (NAcc), depressive-like behavior was elicited, similar to that observed when the deletion was present across the entire brain. In contrast, anxiety levels where unaffected by this localized genetic modification, suggesting that Vnut in astrocytes from the NAcc does not participate in the regulation of anxiety-related behavior [151]. This study is another example, among others, where the authors did not further investigate the mechanisms underlying anxiety regulation by astrocytes.

## 13. Discussion

The therapies currently available for treating mood disorders, including anxiety disorders, show a highly variable success rate, and some of them present non-negligible side effects [2,153,154]. The unsatisfactory efficacy of pharmacological treatments may leave patients in a state of frustration that can worsen their already debilitating conditions and impose significant costs on society. The modest efficacy of pharmacological treatments reveals the presence of a complex and multifactorial etiology in anxiety disorders. The field of oncology has gone a long way from non-specific, aggressive and often unsuccessful chemotherapies to targeted, biological therapies tailored to the unique characteristics of each patient’s tumors. In neuropsychiatric disorders, a similar progress has not yet been achieved [155] and it is urgent that “modern” societies, which frequently promote lifestyles detrimental to the mental health of people from all ages, face this enormous challenge. Setting aside the current technical challenges in manipulating specific cellular populations and brain circuits in humans to achieve targeted interventions, the emphasis now should be on identifying the factors that make individuals more resistant or more vulnerable to everyday anxious stimuli.

In anxiety research, one major challenge is the difficulty of accurately modeling human pathology in animal studies. Despite this limitation, several brain circuits have been identified as mediators of anxiety-like behavior. Given the crucial contributions of astrocytes to higher brain functions, it is critical to include these glial cells in anxiety research. The studies reviewed here suggest that research on astrocytes in anxiety-related behavior is still in its infancy, with only a few initial steps taken. Studies focusing on astrocytes and anxiety behavior in a systematic and in-depth manner are scarce. Instead, many studies investigating astrocytes in other mood disorders give only peripheral attention to their role in anxiety, and even when some effects on anxiety are noted, the underlying mechanisms are often not further explored. Moreover, in some brain regions crucial to anxiety regulation, such as the bed nucleus of the stria terminalis (BNST), the role of astrocytes in this regulation remains completely unexplored. Further detailed studies focusing on astrocytes are essential to better understand the pathogenesis of anxiety disorders and to identify potential novel therapeutic targets.

A caveat that emerges from this review is the low attention given to female biology during anxiety research in the past. Although women are more susceptible to stress-related mental disorders, such as anxiety and depression [156], it is surprising that most of the studies reviewed here were performed in male subjects. A recent study reviewed the existing literature to identify the effects of estrous cycle on the behavior of females in rodent anxiety tests (for a detailed discussion of the effects of cyclical variations in females in anxiety, see [157]). Surprisingly, despite clinical evidence highlighting the higher prevalence of anxiety disorders in women, the overall conclusion of this literature review was that female rodents exhibit lower levels of anxiety compared to males, at least in the behavioral test developed and validated with male rodents. While this may suggest the need to adjust current “male” behavioral test to better assess the emotional states in females, the readout of current animal models may be inefficient in modeling emotions in humans [6]. Taking into account these considerations, an effort must be made to include females in psychiatry and anxiety research in general, and in astrocyte-mediated anxiety research in particular.

The overall picture that arises from this review is rather complex, with coexisting astrocyte-mediated anxiolytic and anxiogenic effects. This complexity reflects the numerous and diverse actions that astrocytes exert on brain function, in cooperation with neurons, in a circuit-dependent manner. To develop alternative treatments for anxiety, we are entrusted with fully exploring the potential that astrocytes offer in modulating brain circuits.

## Figures and Tables

**Figure 1 ijms-26-02774-f001:**
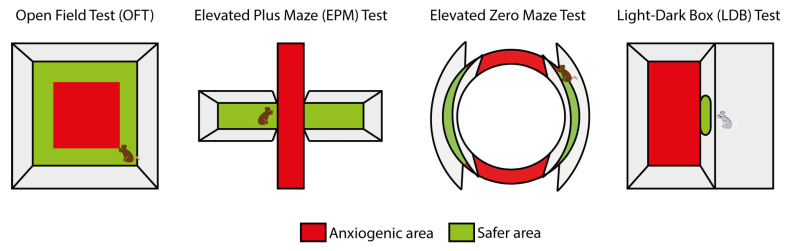
Common tests used to assess changes in anxiety levels in rodents. Open field test (OFT), elevated plus maze (EPM) test, elevated zero maze test and light–dark box (LDB) test. The more anxiogenic areas are shown in red, and the safer areas in green.

**Figure 2 ijms-26-02774-f002:**
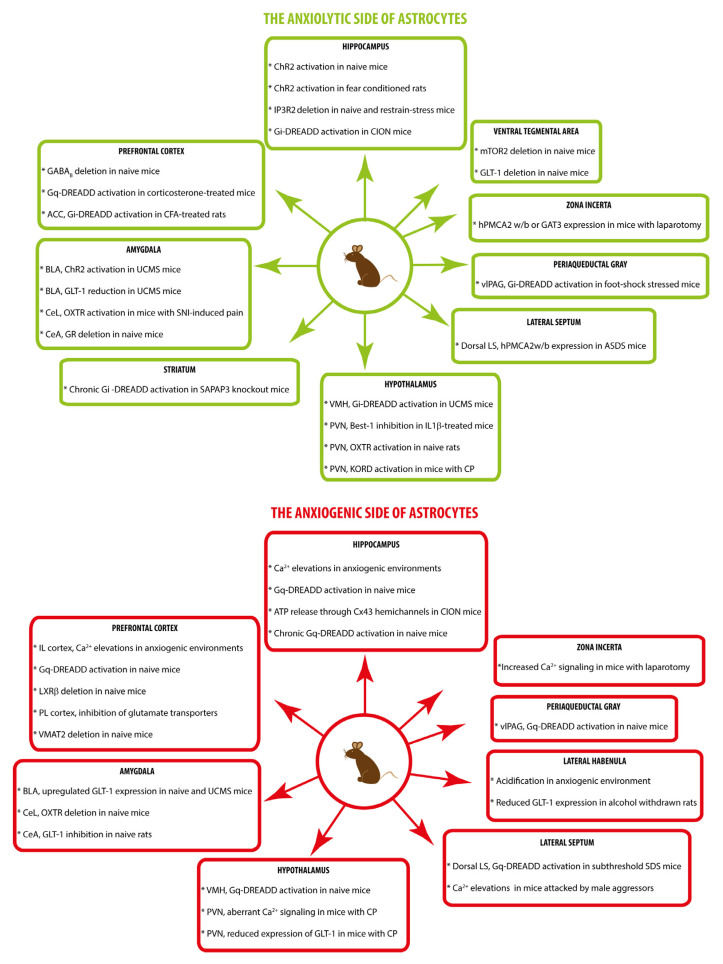
Astrocyte manipulation-mediated anxiolytic/anxiogenic effects and astrocyte characteristics correlated with anxiety in various brain circuits involved in anxiety regulation. For details, see the text.

**Figure 3 ijms-26-02774-f003:**
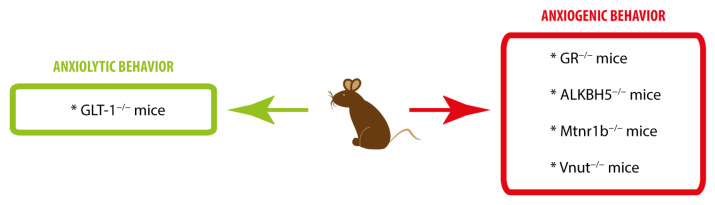
Knockout mice in astrocytes with anxiolytic and anxiogenic phenotypes.

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
