# Peer review of "Astrocytes in Rodent Anxiety-Related Behavior: Role of Calcium and Beyond"

_ijms, 2025, doi:10.3390/ijms26062774_

Round 1

Reviewer 1 Report

Comments and Suggestions for Authors

This review article stands out for its comprehensive and insightful treatment of the role of astrocytes in modulating anxiety-like behaviors in rodent models. The manuscript does an excellent job of synthesizing a large body of evidence that spans multiple brain regions-including the hippocampus, prefrontal cortex, amygdala, striatum, and hypothalamus-to demonstrate how astrocyte signaling, particularly via calcium dynamics, contributes to the regulation of anxiety.

Key strengths include:

- A thorough integration of recent experimental findings: The review meticulously describes both Ca²⁺-dependent and Ca²⁺-independent mechanisms, discussing advanced methods such as optogenetic and chemogenetic approaches. This detail not only enriches our understanding of astrocytic functions but also underscores the importance of methodological nuances in interpreting behavioral outcomes.

- Clear and structured presentation: The article is well organized into sections that address the contributions of astrocytes in distinct brain circuits. Each section effectively links molecular events—such as the activation of specific receptors or signaling pathways—to the resulting behavioral phenomena, providing readers with a logical and accessible narrative.

• Innovative perspective on therapeutic targets: By highlighting the bidirectional control of anxiety through astrocyte activity and signaling pathways, the manuscript opens up exciting avenues for potential therapeutic interventions. The discussion on the modulation of neuronal circuits via astrocyte-derived signals, including ATP and glutamate, is particularly promising for future research into anxiety disorders.

- Balanced and critical analysis: Although the field is complex and sometimes yields contrasting results (for instance, the differing effects observed with various activation protocols), the review acknowledges these nuances with clarity. It thoughtfully discusses the limitations and challenges of exogenous astrocyte activation, thereby providing a balanced perspective that reinforces the credibility of its conclusions.

Overall, the article is an impressive contribution to the field of molecular neuroscience. Its integrative approach not only advances our understanding of the neurobiological underpinnings of anxiety but also paves the way for novel, astrocyte-targeted therapeutic strategies. The clarity, depth, and innovative insights presented in this review make it a valuable resource for both researchers and clinicians interested in the molecular mechanisms of anxiety.

Comments on the Quality of English Language

It's fine, minor errors.

Author Response

Reviewer 1

This review article stands out for its comprehensive and insightful treatment of the role of astrocytes in modulating anxiety-like behaviors in rodent models. The manuscript does an excellent job of synthesizing a large body of evidence that spans multiple brain regions-including the hippocampus, prefrontal cortex, amygdala, striatum, and hypothalamus-to demonstrate how astrocyte signaling, particularly via calcium dynamics, contributes to the regulation of anxiety.

Key strengths include:

- A thorough integration of recent experimental findings: The review meticulously describes both Ca²-dependent and Ca²-independent mechanisms, discussing advanced methods such as optogenetic and chemogenetic approaches. This detail not only enriches our understanding of astrocytic functions but also underscores the importance of methodological nuances in interpreting behavioral outcomes.

- Clear and structured presentation: The article is well organized into sections that address the contributions of astrocytes in distinct brain circuits. Each section effectively links molecular events—such as the activation of specific receptors or signaling pathways—to the resulting behavioral phenomena, providing readers with a logical and accessible narrative.

  •  Innovative perspective on therapeutic targets: By highlighting the bidirectional control of anxiety through astrocyte activity and signaling pathways, the manuscript opens up exciting avenues for potential therapeutic interventions. The discussion on the modulation of neuronal circuits via astrocyte-derived signals, including ATP and glutamate, is particularly promising for future research into anxiety disorders.

- Balanced and critical analysis: Although the field is complex and sometimes yields contrasting results (for instance, the differing effects observed with various activation protocols), the review acknowledges these nuances with clarity. It thoughtfully discusses the limitations and challenges of exogenous astrocyte activation, thereby providing a balanced perspective that reinforces the credibility of its conclusions.

Overall, the article is an impressive contribution to the field of molecular neuroscience. Its integrative approach not only advances our understanding of the neurobiological underpinnings of anxiety but also paves the way for novel, astrocyte-targeted therapeutic strategies. The clarity, depth, and innovative insights presented in this review make it a valuable resource for both researchers and clinicians interested in the molecular mechanisms of anxiety.

Comments on the Quality of English Language

It's fine, minor errors.

I thank the reviewer for the positive comments on the manuscript, particularly for stating that it is “a valuable resource for both researchers and clinicians”.

Reviewer 2 Report

Comments and Suggestions for Authors

IJMS-3475108

The review paper proposed for publication by Gomez-Gonzalo addresses the interesting topic of the contribution of astrocytes in anxiety-like phenotypes. The Author focuses on studies carried out on the preclinical model (rodent, mainly mouse), reporting the results of recent literature subdividing them based on specific brain regions, but also offers insights into genetic models with total deletion.
The manuscript is overall well organized and written, with arguments that can be easily followed by the reader, especially on the basis of an excellent introduction which contains most of the notions useful for understanding the techniques used to collect the data subsequently exposed and discussed.
However, I believe that small changes suggested below could further increase the value of the work for readers.

I would also include the zero maze in Figure 1 for a more complete overview.

I would suggest replacing Figure 2 and 3 with tables so as to allow the reader to synoptically grasp the commonalities and differences of the approaches used for the different structures, possibly with the addition of the tests used to evaluate the behavioral phenotype.

lines 199-202: Please report reference(s) supporting this statement.

lines 515: Please report references supporting the homology/analogy (?) between ACC in human and IL in mouse and, given this parallel, I suggest to do the same for PL (following reported in the same paragraph).

lines 1093-1096 While I agree with the concept expressed, I would not refer to oncology but rather to neuropsychiatric disorders, for which the approach to personalization is however already widely supported with policies and studies.

Furthermore, since as correctly indicated by the Author, locomotor activity can influence anxiety measures, I would suggest specifying in the text, where not already reported, which tests the locomotor activity was measured with.

I also suggest removing the information relating to bone loss in the "hypothalamus" paragraph, which is interesting but possibly distracting for the readers.

I also ask the author to double-check in the test for typos (e.g. line 1014 “OPT”), misspelling (e.g.  line 1048 “antidepressant-like behavior”) and that abbreviation corresponds to the term shown and vice versa.

Author Response

Reviewer 2

The review paper proposed for publication by Gomez-Gonzalo addresses the interesting topic of the contribution of astrocytes in anxiety-like phenotypes. The Author focuses on studies carried out on the preclinical model (rodent, mainly mouse), reporting the results of recent literature subdividing them based on specific brain regions, but also offers insights into genetic models with total deletion.
The manuscript is overall well organized and written, with arguments that can be easily followed by the reader, especially on the basis of an excellent introduction which contains most of the notions useful for understanding the techniques used to collect the data subsequently exposed and discussed.
However, I believe that small changes suggested below could further increase the value of the work for readers.

I thank the reviewer for the positive comments on the manuscript, particularly for stating that “arguments can be easily followed by the reader”.

I would also include the zero maze in Figure 1 for a more complete overview.

As suggested by the reviewer, I have included the zero maze in figure 1.

I would suggest replacing Figure 2 and 3 with tables so as to allow the reader to synoptically grasp the commonalities and differences of the approaches used for the different structures, possibly with the addition of the tests used to evaluate the behavioral phenotype.

I agree with the reviewer that tables help summarize results in Review articles. However, this review is long and, as also noted by Reviewer 3, quite detailed. Therefore, I decided to replace the tables with schematic, color-coded figures to lighten this manuscript. To facilitate comparisons between anxiolytic and anxiogenic astrocyte effects, I have combined the previous Figures 2 and 3 into a single new Figure 2. Hopefully, Reviewer 2 will agree with keeping the new Figure 2, as it also summarizes the key results obtained for each brain structure.

lines 199-202: Please report reference(s) supporting this statement.

As suggested, I have included the reference supporting this statement.

lines 515: Please report references supporting the homology/analogy (?) between ACC in human and IL in mouse and, given this parallel, I suggest to do the same for PL (following reported in the same paragraph).

As suggested by the reviewer, I have included a reference that discusses the homologies of these regions across humans and rodents (new lines 422-423).

lines 1093-1096 While I agree with the concept expressed, I would not refer to oncology but rather to neuropsychiatric disorders, for which the approach to personalization is however already widely supported with policies and studies.

I am not a clinician and may not have an up-to-date perspective on the recent progress made in the treatment of neuropsychiatric disorders that is moving the field forward. However, from my position studying brain mechanisms, my impression is that the intrinsic complexity of brain function, as well as psychiatric disorders, is so vast that achieving personalized therapies tailored to the unique characteristics of each patient’s brain dysfunction remains a significant challenge, at least in our time. This is mainly due to the often unknown etiology of psychiatric disorders in each patient. For this reason, I would like to keep this sentence and add a recent editorial from Nature Mental Health that addresses this important issue.

Furthermore, since as correctly indicated by the Author, locomotor activity can influence anxiety measures, I would suggest specifying in the text, where not already reported, which tests the locomotor activity was measured with.

As suggested by the reviewer, I have included this information throughout the manuscript. I thank the reviewer for this valuable suggestion, which helped identify several studies that lacked this important control.

I also suggest removing the information relating to bone loss in the "hypothalamus" paragraph, which is interesting but possibly distracting for the readers.

As suggested by the reviewer, I have removed the results on bone loss from the hypothalamus section. I thank the reviewer for this suggestion, which helps keep the focus on the subject of the present review.

I also ask the author to double-check in the test for typos (e.g. line 1014 “OPT”), misspelling (e.g.  line 1048 “antidepressant-like behavior”) and that abbreviation corresponds to the term shown and vice versa.

As suggested, I have checked for other typos as well as for abbreviations, including terms such as Channelrhodopsin-2 (ChR2), adeno-associated virus (AAV), short hairpin RNA (shRNA), small interference RNA (siRNA) and monoamine oxidase B (MAOB), among others. I apologize for including abbreviations without their full terms in the previous version of the manuscript.

Reviewer 3 Report

Comments and Suggestions for Authors

The review is quite detailed, but it feels dense at times. The author elaborates on certain findings more than necessary, making it challenging to stay engaged. For example, in lines 557 to 577, it seems like the author is simply describing the results of that paper rather than critically analyzing or summarizing its key contributions, which should be the main goal of a review.

Author Response

Reviewer 3

The review is quite detailed, but it feels dense at times. The author elaborates on certain findings more than necessary, making it challenging to stay engaged. For example, in lines 557 to 577, it seems like the author is simply describing the results of that paper rather than critically analyzing or summarizing its key contributions, which should be the main goal of a review.

The goal of the present review was also to reach an audience unfamiliar with preclinical studies. For this reason, I decided to describe in detail the results of the original studies presented in this review. I apologize if I provided more details than strictly necessary. As suggested by the reviewer 2, I have removed the results on bone loss from the hypothalamus section. This change helps maintain the focus on the subject of the present review. While I agree with the reviewer that lines 557 to 577 in the previous version were mainly descriptive and lack critical comments, in many other sections of the manuscript an effort was made to critically comment on the presented results (see, for example, paragraph 256, lines 639-644, lines 900-902 and lines 1067-1069 in the revised version, among others), particularly emphasizing the results obtained by different groups that are apparently contrasting (see, for example, paragraph 344, lines 529-534 and lines 1010-1022 in the revised version, among others). Moreover, in the Introduction, I critically introduced certain aspects of astrocyte studies that must be considered when analyzing the results obtained in several of the studies presented in this review (see, for example, lines 120 to 136 and paragraph 157 in the revised version, among others). Finally, in the Discussion, I raised some critical concerns that remain in astrocyte studies within the anxiety field, such as the minimal inclusion of females in these studies. Following the reviewer’s criticism, I have reviewed the entire manuscript, critically emphasizing some new aspects (see lines 387-391 and lines 587-589 in the revised version).